# RetCompletion: High-Speed Inference Image Completion with Retentive Network

## Abstract

Time cost is a major challenge in achieving high-quality pluralistic image completion. Recently, the Retentive Network (RetNet) in natural language processing offers a novel approach to this problem with its low-cost inference capabilities. Inspired by this, we apply RetNet to the pluralistic image completion task in computer vision. We present RetCompletion, a two-stage framework. In the first stage, we introduce Bi-RetNet, a bidirectional sequence information fusion model that integrates contextual information from images. During inference, we employ a unidirectional pixel-wise update strategy to restore consistent image structures, achieving both high reconstruction quality and fast inference speed. In the second stage, we use a CNN for low-resolution upsampling to enhance texture details. Experiments on ImageNet and CelebA-HQ demonstrate that our inference speed is $10\times$ faster than ICT and $15\times$ faster than RePaint. The proposed RetCompletion significantly improves inference speed and delivers strong performance, especially when masks cover large areas of the image.

## 1 Introduction

Pluralistic image completion, also known as image inpainting, is a crucial research area with various applications, including object removal and photo restoration Barnes et al. (2009); Criminisi et al. (2004); Dale et al. (2009); Wan et al. (2020). CNN-based methods **?**Iizuka et al. (2017); Li et al. (2017) have demonstrated impressive results by capturing local texture patterns, but they often struggle to model global structures, leading to suboptimal image reconstruction quality. To overcome this limitation, researchers have introduced hybrid models combining Transformers and CNNs Wan et al. (2021); Zheng et al. (2022); Li et al. (2022). While these approaches significantly improve reconstruction quality and produce diverse results by modeling the underlying data distribution, Transformer-based pixel-wise generation involves extensive feature fusion calculations. This computational overhead increases inference time, limiting the practicality of these methods, especially in real-time applications. Therefore, developing algorithms that maintain high-quality reconstruction while improving computational efficiency remains a critical challenge in this domain.

Recently, Retentive Network (RetNet) Sun et al. (2023) has shown substantial potential in natural language processing due to its multi-scale retention mechanism, which bridges parallel training and recurrent inference. This capability enables RetNet to process information efficiently, even in pixel-wise generation tasks. However, applying RetNet directly to vision tasks presents challenges, as image information is not unidirectional like language data.

In this work, we propose RetCompletion, a novel image completion framework designed to address the challenges of slow inference and inconsistent image reconstruction. RetCompletion operates in two stages: the first stage leverages a Bi-RetNet architecture for low-resolution pixel-wise image generation, while the second stage uses a CNN for high-resolution texture refinement. Extensive experiments on datasets such as ImageNet and CelebA-HQ demonstrate that RetCompletion significantly accelerates inference while maintaining high reconstruction quality.

The key contributions of this work are:

1. **First application of RetNet to image completion**: We introduce RetNet for the first time in image completion tasks, utilizing its parallel training and recursive inference to accelerate the process.

2. **Bi-RetNet with bidirectional fusion**: Our Bi-RetNet architecture fuses forward and backward contextual information, improving consistency and realism, particularly when reconstructing large masked areas.

3. **Efficient pixel-wise inference based on RetNet**: RetCompletion's pixel-wise inference strategy, enabled by RetNet, is significantly faster than Transformer-based methods and produces better overall results by incorporating previously generated pixel information during inference.

## 2  RELATED WORK

**Pluralistic Image Completion**  The significance of Pluralistic Image Completion lies in providing a diverse approach to image processing, allowing for the creation of images with different styles and effects, enriching the toolbox in creative and design fields, and supporting diverse choices in decision-making processes. PIC Zheng et al. (2019) employs a dual-path framework based on probabilistic principles: one is the reconstructive path, which utilizes the given ground truth to obtain prior information about the missing parts and reconstructs the original image from this distribution. The other is the generative path, where the conditional prior is coupled with the distribution from the reconstructive path. ICT Wan et al. (2021) directly optimizes the log-likelihood in the discrete space in the first transformer-based stage without the need for additional assumptions. RePaint Lugmayr et al. (2022) applies the Diffusion model to the image inpainting task, using a pre-trained unconditional DDPM Ho et al. (2020) as the generative prior and modifying the reverse diffusion iterations by sampling the unmasked regions from the given image information. Since this technique doesn't alter or condition the original DDPM Ho et al. (2020) network itself, the model can generate high-quality and diverse output images for any inpainting scenario.

**Retentive Network**  Retentive Network Sun et al. (2023) introduces the retention mechanism with a dual form of recurrence and parallelism. It has three computation paradigms,i.e., parallel, recurrent, and chunkwise recurrent. We can train parallelly by using parallel paradigm while conducting inference recurrently using recurrent and chunkwise paradigms. The retention mechanism utilizes a rotation-based positional encoding along with a decay term to effectively model the position information of tokens, known as xPos Sun et al. (2022), a relative position embedding proposed for Transformer. We attempt to extend this method to two-dimensional images.

## 3  METHODS

The overall pipeline of our method can be seen in Figure. 1, which consists of two stages. The first stage is utilized to complete low-resolution images based on Bi-RetNet, while the second stage generates high-resolution images based on CNN.

### 3.1  RETENTIVE NETWORK

Retentive Network (RetNet) is a powerful architecture initially designed for natural language processing, which combines parallel and recurrent representations to efficiently handle sequential data. Its key advantage lies in its ability to balance parallel training and recurrent inference, enabling fast computation even in complex tasks.

RetNet models sequences in a recurrent manner, where the hidden state at each step is computed as:

$$s_n = As_{n-1} + K_n^\top v_n \tag{1}$$

This allows RetNet to accumulate information over time while maintaining efficient updates.

Additionally, RetNet can be diagonalized to simplify the recurrence into a more efficient form using positional encoding, further improving its speed and accuracy:

$$o_n = \sum_{m=1}^{n} \gamma^{n-m}(Q_n e^{in\theta})(K_m e^{im\theta})^\dagger v_m \tag{2}$$

The RetNet framework incorporates three key representations:

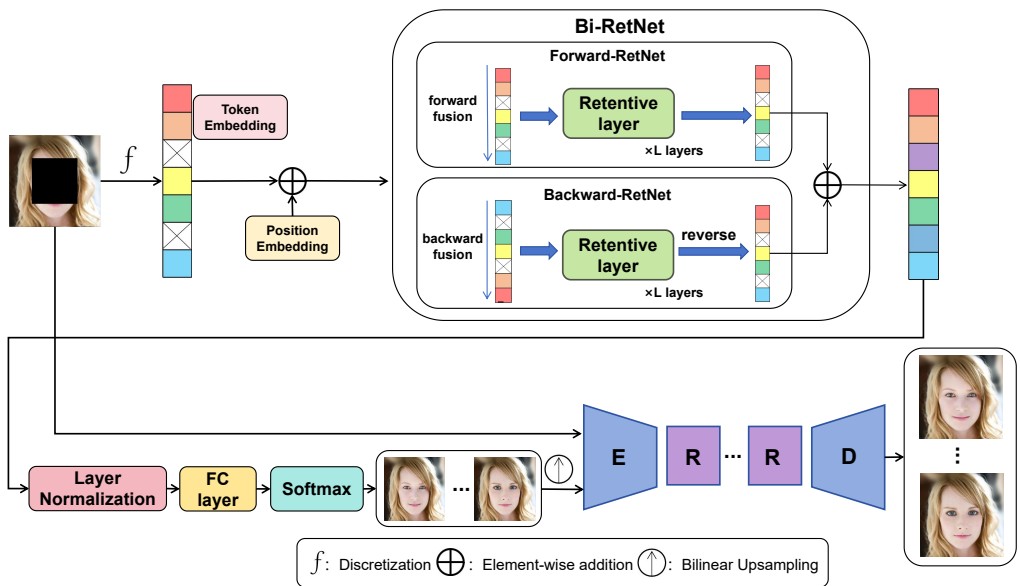

Figure 1: **Pipeline Overview.** Our method consists of two networks, which are trained separately. Based on the Bi-RetNet, the first network is employed for completing low-dimensional images. A parallel representation is utilized during training, predicting all pixels simultaneously to expedite the training process. In contrast, during inference, a recurrent representation is employed, predicting one pixel at a time to enhance the quality of the generated image. The second network, built on a CNN architecture, comprises an encoder, a decoder, and multiple residual blocks. Its primary function is to restore high-dimensional images from their low-dimensional counterparts.

**Parallel Representation**    Parallelization allows RetNet to achieve linear complexity during training, greatly speeding up the process. This is computed as:

$$Retention(X) = (QK^T \odot D)V \tag{3}$$

**Recurrent Representation**    During inference, RetNet uses its recurrent form, which achieves O(1) complexity, enabling fast pixel-wise inference:

$$Retention(X_n) = Q_n S_n \tag{4}$$

**Chunkwise Recurrent Representation**    This combines both parallel and recurrent approaches, allowing for accelerated computations in large-scale tasks:

$$Retention(X_{[i]}) = (Q_{[i]} K_{[i]}^T \odot D)V_{[i]} + Q_{[i]} R_{[i-1]} \tag{5}$$

This unique combination of representations allows RetNet to efficiently process large datasets, making it particularly well-suited for pixel-wise image completion tasks, where both speed and accuracy are critical.

### 3.2 PREPROCESSING

To reduce the computational cost of attention calculations during preprocessing, we first downsample the input image from its original resolution $H \times W$ to a lower-resolution version $L \times L$:

$$\bar{I}L \times L \times 3 = \text{Downsampling}(IH \times W \times 3) \tag{6}$$

This step simplifies the image, reducing the number of pixels that need to be processed during subsequent stages.

RGB color channels typically exist in a high-dimensional space ($256^3$ colors), which makes direct processing computationally expensive. To handle this, we generate a compact visual vocabulary by applying K-Means clustering on the entire ImageNet dataset Russakovsky et al. (2015), reducing the space to 512 representative colors. For each unmasked pixel, we map its color to the nearest representative color from this vocabulary. The image is then raster-scanned and reshaped into a sequence, which is necessary for the RetNet model:

$$S_{L^2 \times 1} = \text{reshape}(\text{project}(\bar{I}_{L \times L \times 3})) \tag{7}$$

We also create a binary mask sequence, where 1 indicates a masked pixel and 0 represents an unmasked pixel:

$$M_{L^2 \times 1} = \mathbb{I}(S_i \text{ is masked}), i = 1, 2, \ldots, L^2 \tag{8}$$

**Feature Encoding**  To convert each pixel's color into a feature vector, we use a trainable embedding. This transforms the discrete color values from the visual vocabulary into $d$-dimensional feature vectors, which will serve as inputs to RetNet.

**Position Encoding**  RetNet uses positional encoding to track the location of tokens in a sequence. For 2D images, we introduce a learnable position embedding that captures spatial information. This embedding, combined with the feature encoding, forms the input sequence for RetNet:

$$X_{L^2 \times d} = \text{FE}(S_{L^2 \times 1} \odot M_{L^2 \times 1}) + \text{PE}_{L^2 \times d} \tag{9}$$

By combining the color features and positional information, this sequence serves as the input to the RetNet model, allowing it to process the image efficiently in the subsequent stages.

### 3.3 APPEARANCE PRIORS RECONSTRUCTION BY BI-RETNET

Traditional RetNet operates with unidirectional information flow, which is suitable for natural language processing. However, image completion requires integrating contextual information from multiple directions. To address this, we developed a bidirectional RetNet model consisting of a Multi-Head Forward-RetNet and a Multi-Head Backward-RetNet. These two RetNets share the same structure but have different parameters, enabling them to capture information from different directions.

**Multi-Head Forward-RetNet**  In this model, we utilize $h$ heads, where each head has a feature dimension of $d_{\text{head}} = d/h$. Different heads use different parameter matrices $W_Q, W_K, W_V \in \mathbb{R}^{d_{\text{head}} \times d_{\text{head}}}$. The retention for each head is computed as:

$$head_i = Retention(X, W_i) \tag{10}$$

The multi-head outputs are concatenated and normalized using GroupNorm:

$$Y = GroupNorm_h(Concat(head_1, \ldots, head_h)) \tag{11}$$

The final output for each forward layer is computed as:

$$Y_{forward}^l = MSR(LN(X_{forward}^l)) + X_{forward}^l \tag{12}$$
$$X^{l+1}forward = FFN(LN(Y^l forward)) + Y_{forward}^l \tag{13}$$

where $X^1$ is the sequence obtained from the preprocessing stage, and $l$ denotes the layer index.

**Multi-Head Backward-RetNet**  The Multi-Head Backward-RetNet follows the same procedure, except that it processes the reversed input sequence. After computation, the result is reversed back to its original order, yielding $X_{backward}^{(L+1)}$.

**Feature Fusion** To combine the forward and backward information, we perform feature fusion by adding the outputs from the forward and backward passes. Layer normalization, fully connected layers, and softmax are then applied to produce a per-pixel distribution of 512 possible colors:

$$P(x|X,\theta) = softmax(FC(LN(X^{L+1}forward + X^{L+1}backward))) \tag{14}$$

This fusion of forward and backward information allows the model to capture richer contextual dependencies, leading to more accurate and coherent image reconstructions.

**Loss Function** Similar to BERT Devlin et al. (2018), we employ the Masked Language Model (MLM) objective to optimize the RetNet. The loss function minimizes the negative log-likelihood of the masked pixels, ensuring that the model learns to predict missing regions accurately:

$$L_{\text{MLM}} = \mathbb{E}_{X}[\frac{1}{N} \sum_{n=1}^{N} -\log p(x_n|X,\theta)] \tag{15}$$

where $N$ represents the number of masked pixels in the image. By minimizing this loss, the generated images approach the ground truth, resulting in high-quality reconstructions.

## 3.4 Parallel Training

During training, we utilize both the parallel and chunkwise recurrent representations to accelerate the process. Specifically, we choose to predict all masked pixels simultaneously, rather than sequentially, in order to improve training efficiency.

In this approach, masked pixels receive color information exclusively from unmasked pixels, meaning that masked pixels do not incorporate information predicted for other masked pixels, even those earlier in the sequence. This strategy allows for more efficient computation, as it reduces dependencies between predictions and enables faster iteration over large datasets.

By employing this parallelized method, we are able to reduce the overall training time, especially when handling high-dimensional data.

## 3.5 Pixel-wise Inference

In the inference stage, we adopt a pixel-wise inference method, which has proven to be significantly more effective compared to predicting all pixels simultaneously. The pixel-wise approach allows the model to incorporate newly predicted pixel information step by step, improving the overall quality of the generated images. This advantage is made possible by the Bi-RetNet architecture, which enables fast updates during inference, a capability that Transformer-based models lack.

We begin by performing information fusion on the initial image to generate integrated representations, $S_{forward}$ and $S_{backward}$. Then, we predict the masked pixels one by one in a raster-scan manner. At each step, we update the retention state of the forward RetNet with the new pixel information, ensuring that each subsequent pixel prediction benefits from previous predictions.

The inference process is detailed in Algorithm 1, where the model integrates the forward and backward information for each pixel prediction and updates the RetNet's retention state after each step:

This pixel-wise inference strategy allows our model to efficiently update and refine each pixel prediction, improving both speed and accuracy compared to Transformer-based methods.

## 3.6 Guided Upsampling

After reconstructing the appearance priors, we reshape the sequence $X \in \mathbb{R}^{L^2 \times 3}$ into $I_t \in \mathbb{R}^{L \times L \times 3}$, which represents a low-resolution image. We then upscale this image to the original resolution $H \times W \times 3$. Following ICT Wan et al. (2021), we employ a CNN-based guided upsampling network, as CNNs have shown excellent performance in texture reconstruction. The upsampling network is composed of an encoder, decoder, and residual blocks. First, we upsample $I_t$ to the original resolution

---

**Algorithm 1** Pixel-wise Inference

---

1: **Initialization:**
2: Compute initial $Q_0 = X_0 W_Q$, $K_0 = X_0 W_K$, $V_0 = X_0 W_V$
3: Initialize $S_{forward}$ and $S_{backward}$ with $Q_0, K_0, V_0$
4: **Pixel-wise Inference:**
5: **for** $n = 1$ to $N$ (where $j_n$ are the indices of masked pixels) **do**
6:     Retrieve positional encodings: $\bar{X}_{forward,n1} = PE_{j_n}$, $\bar{X}_{backward,n1} = PE_{j_n}$
7:     **for** each layer $l$ from 1 to $L$ **do**
8:         # Forward Pass
9:         $Y_{forward,nl} = MSR(LN(\bar{X}_{forward,nl})) + \bar{X}_{forward,nl}$
10:         $\bar{X}_{forward,n(l+1)} = FFN(LN(Y_{forward,nl})) + Y_{forward,nl}$
11:         # Backward Pass
12:         $Y_{backward,nl} = MSR(LN(\bar{X}_{backward,nl})) + \bar{X}_{backward,nl}$
13:         $\bar{X}_{backward,n(l+1)} = FFN(LN(Y_{backward,nl})) + Y_{backward,nl}$
14:     **end for**
15:     # Combine forward and backward results for prediction
16:     $P(x_n) = \text{softmax}(FC(LN(\bar{X}_{forward,n(L+1)} + \bar{X}_{backward,n(L+1)})))$
17:     Sample pixel value: $x_n \sim P(x_n)$
18:     Update pixel embedding: $X_n = FE(x_n) + PE_{j_n}$
19:     # Update forward RetNet state
20:     Compute new $Q_n = X_n W_Q$, $K_n = X_n W_K$, $V_n = X_n W_V$
21:     Update $S_{forward}$ with $Q_n, K_n, V_n$
22: **end for**

---

using bilinear interpolation, and then we feed the upsampled image along with the original image and mask into the upsampling network as:

$$I_{\text{pred}} = F_\delta(I_t^\uparrow \frown I_m) \in \mathbb{R}^{H \times W \times 3} \tag{16}$$

where $F$ represents the upsampling network with parameters $\delta$.

We apply both $L_1$ loss between $I_{\text{pred}}$ and $I$, and adversarial loss to train the upsampling network as:

$$L_{L_1} = \mathbb{E}[|I_{\text{pred}} - I|_1] \tag{17}$$

$$L_{\text{adv}} = \mathbb{E}[\log(1 - D\omega(I\text{pred}))] + \mathbb{E}[\log D_\omega(I)] \tag{18}$$

where $D$ is the discriminator with parameters $\omega$.

The upsampling network $F$ and discriminator $D$ are trained with the following optimization objective:

$$\min_F \max_D L_{\text{upsample}}(\delta, \omega) = \alpha_1 L_{L_1} + \alpha_2 L_{\text{adv}} \tag{19}$$

## 4 EXPERIMENTS

In our experiments, we evaluate the performance of the proposed method using two datasets: CelebA-HQ Karras et al. (2017) and ImageNet Russakovsky et al. (2015). We perform both quantitative and qualitative evaluations to assess the quality of image completion and the inference speed. Quantitative comparisons are conducted against other state-of-the-art methods in terms of image quality and computational efficiency, while qualitative comparisons are based on user feedback. Note that all qualitative and quantitative results reported in this paper are based on a fixed image resolution of 256 pixels.

| Datasets | h | d | N | $\mathbb{L}$ |
|---|---|---|---|---|
| CelebA-HQ Karras et al. (2017) | 8 | 512 | 30 | $48\times48$ |
| ImageNet Russakovsky et al. (2015) | 8 | 1024 | 35 | $32\times32$ |

Table 1: **Retention Network parameter setting across different experiment**. h: Head number. d: The dimension of embedding space. N: Number of retention layers. **L**: The length of appearance prior.

### 4.1 IMPLEMENTATION DETAILS

To ensure fair comparisons across different datasets and methods, we follow the same configuration as ICT Wan et al. (2021) for all experiments, as shown in Table 1. For the CelebA-HQ Karras et al. (2017) and ImageNet Russakovsky et al. (2015) datasets, we retain the original training and test splits. Additionally, we employ PConv Liu et al. (2018) to generate diverse masks during training to simulate various image occlusion scenarios.

### 4.2 QUANTITATIVE COMPARISONS

We quantitatively compare our method against several state-of-the-art (SOTA) image completion techniques using peak signal-to-noise ratio (PSNR), structural similarity index (SSIM), and learned perceptual image patch similarity (LPIPS). Experiments are conducted on CelebA-HQ Karras et al. (2017) and ImageNet Russakovsky et al. (2015) datasets with five distinct mask types to assess performance across different occlusion patterns. For all pluralistic image completion methods, Top-1 sampling is applied during testing.

The results of the quantitative experiments are presented in Table 2. It is clear that our method consistently outperforms most existing SOTA methods across various mask types and datasets. A notable observation is the significant difference between pixel-wise inference and simultaneous pixel estimation (denoted with * in the table). Our pixel-wise inference approach demonstrates clear superiority in terms of both image quality and perceptual similarity, as evidenced by the improvements in PSNR, SSIM, and LPIPS metrics. This emphasizes the effectiveness of our method, particularly in scenarios with complex occlusions.

While this section highlights the quality improvements, the efficiency of our method is also noteworthy. Our pixel-wise inference strategy not only yields better image reconstruction but also achieves faster inference times compared to methods that predict all pixels simultaneously. We will explore this speed advantage in more detail in the subsequent section, where we provide a visual comparison of the inference time between our method and others.

**Mask types**   We utilize five distinct mask types in our experiments. The Wide and Narrow masks are adapted from LaMa Suvorov et al. (2022), representing occlusions of different widths across the image. The Half mask randomly occludes either the top, bottom, left, or right part of the image. The Center mask covers a central $64 \times 64$ region, while the Expand mask occludes all regions except for the central part, covering the majority of the image.

### 4.3 USER STUDY

To enhance the assessment of subjective quality, we additionally perform a user study to compare our method against other baseline approaches. We randomly select 50 images and apply various masks to each. Employing different image completion methods, including pluralistic image completion methods, we consistently used the Top-1 sampling result. Specifically, we present a set of five images generated by MED Liu et al. (2020), PIC Zheng et al. (2019), EC Nazeri et al. (2019), ICT Wan et al. (2021), and our method for each image. Users are then asked to rank the top three images that appear most realistic. Finally, we calculate the percentage of times each method ranked within the top three. Sample images for user study are shown in Figure. 2.

The results obtained from 200 users are shown in Figure 3a. The results show that our method significantly outperforms the PIC method in terms of visual quality. Additionally, our method shows

| Dataset | | CelebA-HQ Karras et al. (2019) | | | ImageNetRussakovsky et al. (2015) | | |
|---|---|---|---|---|---|---|---|
| Method | Mask Ratio | PSNR | SSIM | LPIPS | PSNR | SSIM | LPIPS |
| PIC Zheng et al. (2019) | | 23.781 | 0.883 | 0.164 | 23.765 | 0.819 | 0.234 |
| LaMa Suvorov et al. (2022) | | 27.581 | 0.928 | 0.045 | 26.099 | 0.865 | 0.105 |
| RePaint Lugmayr et al. (2022) | | 27.496 | 0.931 | 0.059 | 25.768 | 0.859 | 0.134 |
| ICT* *Wan et al.* (2021) | Wide | 26.897 | 0.922 | 0.069 | 25.545 | 0.848 | 0.125 |
| ICT Wan et al. (2021) | | 27.139 | 0.932 | 0.063 | 25.886 | 0.862 | 0.107 |
| Ours* | | 27.643 | 0.926 | 0.053 | 25.989 | 0.852 | 0.118 |
| Ours | | 27.966 | 0.938 | 0.042 | 26.087 | 0.869 | 0.103 |
| PIC Zheng et al. (2019) | | 25.823 | 0.901 | 0.062 | 24.091 | 0.823 | 0.098 |
| LaMa Suvorov et al. (2022) | | 28.684 | 0.942 | 0.028 | 26.892 | 0.902 | 0.061 |
| RePaint Lugmayr et al. (2022) | | 28.547 | 0.938 | 0.028 | 26.908 | 0.906 | 0.064 |
| ICT* *Wan et al.* (2021) | Narrow | 28.242 | 0.932 | 0.041 | 26.887 | 0.898 | 0.079 |
| ICT Wan et al. (2021) | | 28.551 | 0.944 | 0.036 | 26.902 | 0.903 | 0.073 |
| Ours* | | 28.397 | 0.935 | 0.031 | 26.882 | 0.901 | 0.071 |
| Ours | | 28.692 | 0.943 | 0.029 | 26.911 | 0.902 | 0.065 |
| PIC Zheng et al. (2019) | | 21.484 | 0.852 | 0.238 | 19.498 | 0.708 | 0.354 |
| LaMa Suvorov et al. (2022) | | 25.208 | 0.905 | 0.138 | 23.513 | 0.756 | 0.254 |
| RePaint Lugmayr et al. (2022) | | 24.846 | 0.902 | 0.165 | 23.498 | 0.762 | 0.304 |
| ICT* *Wan et al.* (2021) | Half | 24.356 | 0.896 | 0.179 | 23.476 | 0.748 | 0.278 |
| ICT Wan et al. (2021) | | 24.798 | 0.906 | 0.166 | 23.502 | 0.753 | 0.255 |
| Ours* | | 24.798 | 0.898 | 0.153 | 23.496 | 0.746 | 0.278 |
| Ours | | 25.103 | 0.907 | 0.145 | 23.512 | 0.759 | 0.262 |
| PIC Zheng et al. (2019) | | 25.580 | 0.887 | 0.153 | 23.806 | 0.816 | 0.167 |
| LaMa Suvorov et al. (2022) | | 28.529 | 0.940 | 0.039 | 26.276 | 0.886 | 0.086 |
| RePaint Lugmayr et al. (2022) | | 28.556 | 0.940 | 0.041 | 26.304 | 0.886 | 0.093 |
| ICT* *Wan et al.* (2021) | Center | 28.409 | 0.935 | 0.058 | 26.198 | 0.879 | 0.103 |
| ICT Wan et al. (2021) | | 28.496 | 0.942 | 0.052 | 26.282 | 0.888 | 0.096 |
| Ours* | | 28.504 | 0.932 | 0.045 | 26.245 | 0.880 | 0.092 |
| Ours | | 28.559 | 0.938 | 0.037 | 26.311 | 0.890 | 0.083 |
| PIC Zheng et al. (2019) | | 18.893 | 0.798 | 0.576 | 17.364 | 0.652 | 0.712 |
| LaMa Suvorov et al. (2022) | | 23.382 | 0.878 | 0.342 | 20.384 | 0.697 | 0.534 |
| RePaint Lugmayr et al. (2022) | | 23.376 | 0.882 | 0.435 | 20.439 | 0.702 | 0.629 |
| ICT* *Wan et al.* (2021) | Expand | 23.298 | 0.876 | 0.446 | 20.126 | 0.683 | 0.562 |
| ICT Wan et al. (2021) | | 23.379 | 0.879 | 0.432 | 20.324 | 0.698 | 0.544 |
| Ours* | | 23.339 | 0.872 | 0.398 | 20.218 | 0.696 | 0.541 |
| Ours | | 23.380 | 0.881 | 0.372 | 20.423 | 0.706 | 0.536 |

Table 2: **Quantitative results on CelebA-HQ Karras et al. (2017) and ImageNet Russakovsky et al. (2015) datasets with different mask types.** All the pluralistic image completion methods use Top-1 sampling. The models with * indicate the prediction method that uses all pixel points simultaneously, while the models without * indicate the prediction method that uses pixel-by-pixel prediction.

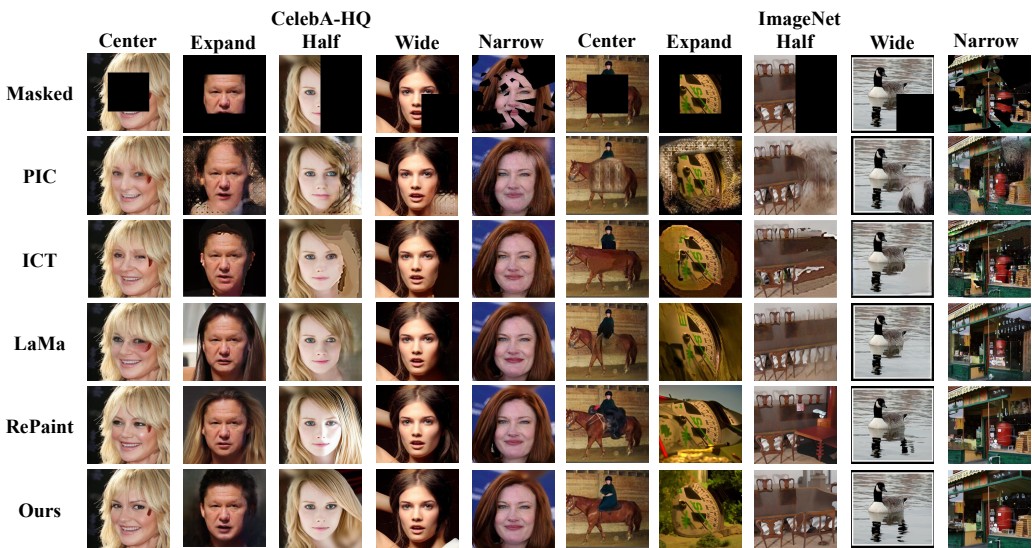

Figure 2: **Sample images for user study.**

a slight advantage over the ICTWan et al. (2021), LaMa Suvorov et al. (2022), and RePaintLugmayr et al. (2022) methods. This demonstrates the superiority of our method in terms of visual perception.

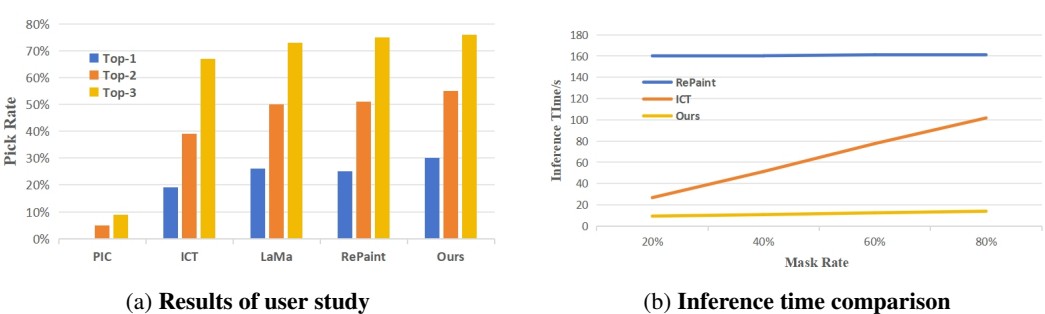

(a) **Results of user study**

(b) **Inference time comparison**

Figure 3: Comparison of user study results and inference time.

## 4.4 INFERENCE TIME

To ensure a fair comparison, we measured the pixel-wise completion inference time of ICT Wan et al. (2021), RePaint Lugmayr et al. (2022), and our proposed method on the ImageNet Russakovsky et al. (2015) dataset using a GeForce RTX 4090 GPU. The results, shown in Figure 3b, demonstrate that our method achieves significantly lower inference times compared to the other two methods, particularly at higher mask rates.

For the RePaint method, although its inference time remains consistent regardless of the mask rate, it requires numerous iterative denoising steps, leading to a substantial overall inference time. On the other hand, ICT Wan et al. (2021) recalculates attention for each pixel estimation, causing the inference time to increase linearly as the mask rate rises. In contrast, our method leverages a recurrent computation paradigm, where only the information of changed pixels is updated and integrated into the state $S$. As a result, our method shows only a minimal increase in inference time across different mask rates, ensuring consistently high-speed performance.

## 5 LIMITATIONS

As illustrated in Figure 2, current image completion methods encounter significant difficulties when dealing with more challenging mask types. For example, the performance on Expand-type masks is particularly subpar, especially when applied to highly diverse datasets such as ImageNet. This highlights a notable gap in the effectiveness of existing approaches. Recognizing this limitation, our future research will place a stronger emphasis on improving the robustness and accuracy of image completion techniques for these complex mask scenarios.

## 6 CONCLUSION

We propose RetCompletion, a two-stage method for pluralistic image completion with three key innovations. First, RetNet is applied for the first time in image completion, offering efficient parallel training and recursive inference. Second, we introduce Bi-RetNet, which integrates bidirectional contextual information to enhance image consistency and reconstruction quality. Third, our pixel-wise inference approach significantly reduces inference time, outperforming Transformer-based methods in computational efficiency. Experiments demonstrate that RetCompletion delivers superior image quality over CNN-based methods and achieves comparable results to Transformer approaches, while maintaining faster inference, making it highly suitable for real-time applications.

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
