# OpenReview forum: "RetCompletion:High-Speed Inference Image Completion with Retentive Network"
_ICLR.cc/2025/Conference — ICLR 2025 Conference Withdrawn Submission_

### Official Review · Reviewer_L4gK · 2024-11-02

**Soundness:** 2
**Presentation:** 2
**Contribution:** 2
**Rating:** 3
**Confidence:** 3

**Summary:**

This paper introduces the Retentive Network named RetCompletion for high-quality image completion, with the goal of reducing the time cost. RetCompletion includes sequence information fusion model that integrates contextual information from images and low-resolution upsampling CNN which enhances texture details.

**Strengths:**

1. A new method for image completion which achieves significant improvement on inference speed compared to previous methods like ICT and Repaint.
2. The paper is well written and the main idea is stated clearly.

**Weaknesses:**

1. The novelty is limited. The main claimed contribution is that the paper first applies RetNet for image completion, which is not sufficient since RetNet has been proposed in NLP.
2. The paper should be reorganized. There is much wasted space in current version like Page 8 and figure 1
3. The authors should provide more visual results. In figure 2, the difference between the proposed method and Repaint on human face cannot be clearly seen.

**Questions:**

The paper should be reorganized because of much wasted space.

---

### Official Review · Reviewer_Es15 · 2024-11-02

**Soundness:** 1
**Presentation:** 2
**Contribution:** 1
**Rating:** 3
**Confidence:** 5

**Summary:**

This paper studies the task of image completion.
The key idea is to use the Retentive Netwotk (RetNet) in natural language processing to this image completion task.
To achieve this goal, a two-stage framework is introduced: 1) a Bi-RetNet network is applied to infer the coarse semantic information from partial visible information. 2) a CNN-based network is built to refine the visual appearance for high-resolution images.
Experiments are conducted on two traditional datasets (CelebA-HQ, ImageNet) and demonstrate reasonable results on image completion.

**Strengths:**

### Compelling results on an interesting task

- The task of generating photorealistic images from partial visible images is an interesting task. The proposed method seems to work well on various masks.
-  Based on the visual results shown in Figure 2, the images are reasonable with better visual appearance. However, they only compared with very old methods.

**Weaknesses:**

### W1 -- Limited contribution of the proposed framework
- The big weakness of the paper is the proposed framework is too similar to existing ICT and TFill, by only replacing the transform with the new Bi-RetNet. The motivation is necessary to be discussed between the proposed method and the existing ICT and TFill. It is hard for me to buy the motivation that RetNet is good in NLP, and we must use it in CV.
- Despite all the efforts in designing the new pipeline that apply natural language processing's architecture into computer vision, the improved performance is limited to prior approaches (and they are old state-of-the-art). Hence, if the authors want to introduce a new architecture from NLP to CV, it would be better to demonstrate the big improvement for the traditional and interesting task.
- L163-L165: K-Means clustering will lose many high-frequency information, which is not the best way to encode the image. Why not use the codebook for discrete representation or the Gaussian distribution for continues representation in Latent diffusion model?

### W2 -- Presentation
- It took me a hard time to actually get the main contribution of this paper, which is hidden in a bunch of overwhelming technical details. Most technical details come from existing approaches, and I cannot figure out what's the main part proposed by this paper.
- L28-33: "pluralistic image completion" is only one direction of image inpainting. The **pluralistic image completion** is proposed by Zheng et al. 2019 for multiple and diverse solutions given a partial visible image. However, many related works mentioned here are not for this new pluralistic image completion task. The authors should clearly distinguish them.
- If the authors start with "pluralistic image completion", the multiple and diverse results are expected in the results. However, I have not found one of them. If the authors do the deterministic solution, PIC and ICT are not good baseline. The better baseline for deterministic results includes TFill (Zheng et al. 2022).
- The quantitative results in Table 2 are also not obvious. The paper would be stronger to highlight the best results, while the improvement is limited in the number.

### W3 -- Clarifications, typos & suggestions
- L29-L30: "CNN-based methods?", the citation is wrong.
- L158-L160 eq. 6. what's the representation of $\hat{I}$ and $I$?
- L298-L300, the merge sign is wrong in eq. 16. Actually, most equations in the paper need to be rewritten and improved.
- L348-L350, "It is clear that..." should be "It is clear demonstrate that..."

**Questions:**

1. What's the difference between the proposed architecture to the TFill (Zheng et al. 2022), which also used a two-stage framework for image completion? The only difference for me is to replace the transform with the RetNet, but this is not a big contribution to the community, and the authors do not demonstrate its effective.
2. The paper would be stronger to compare with the latest state-of-the-art work, instead of the work from 2022. For example, StrDiffusion [r1] and InpaintAnything [r2].

[r1] Liu, H., Wang, Y., Qian, B., Wang, M., & Rui, Y. (2024). Structure Matters: Tackling the Semantic Discrepancy in Diffusion Models for Image Inpainting. In Proceedings of the IEEE/CVF Conference on Computer Vision and Pattern Recognition (pp. 8038-8047).
[r2] Yu, T., Feng, R., Feng, R., Liu, J., Jin, X., Zeng, W., & Chen, Z. (2023). Inpaint anything: Segment anything meets image inpainting. arXiv preprint arXiv:2304.06790.

---

### Official Review · Reviewer_jpMq · 2024-11-04

**Soundness:** 2
**Presentation:** 2
**Contribution:** 2
**Rating:** 3
**Confidence:** 4

**Summary:**

The paper presents a novel application of RetNet in pluralistic image completion. By employing a bidirectional RetNet structure, it effectively adapts the RetNet from the NLP domain to computer vision tasks. Additionally, it utilizes the powerful texture reconstruction capabilities of CNNs to up sample the completed images, restoring them to high-definition quality. The quantitative and qualitative evaluations demonstrate that the network outperforms most existing SOTA methods while also achieving very fast inference speeds.

**Strengths:**

1. The paper quantitatively and qualitatively compares the network with current state-of-the-art models, achieving notable advantages in both evaluations.
2. The network employs a bidirectional RetNet structure, effectively applying RetNet to image tasks without significantly increasing the computational burden.

**Weaknesses:**

1. The paper presents a network structure that appears similar to that in the 'High-fidelity pluralistic image completion with transformers' paper, and the bidirectional model structure shares similarities with Vision Mamba. This raises questions about the novelty of the proposed approach.
2. The readability and formatting of the equations could be improved. The RetNet-related equations lack explanations for the symbols, which affects the readers' understanding. In equation (6), the shape of the image should be in subscript form.
3. While the paper provides a comparison with several established models, it could be strengthened by including more recent SOTA models, such as DiffIR, to provide a more comprehensive evaluation.

**Questions:**

Please refer to the weaknesses.

---

### Official Review · Reviewer_65g5 · 2024-11-04

**Soundness:** 3
**Presentation:** 3
**Contribution:** 2
**Rating:** 6
**Confidence:** 3

**Summary:**

This paper introduces a novel image completion approach using Bi-RetNet, a modified RetNet model originally designed for NLP, to restore low-resolution images, followed by a CNN-based upsampling to produce high-resolution reconstructions. Experiments demonstrate the effectiveness of this approach, achieving both fast and high-quality image completion.

The RetCompletion architecture and pixel-wise inference method are well explained, providing clarity on how it achieves fast inference speeds. The paper includes quantitative, qualitative, and inference time results, with Figure 2 clearly illustrating inpainting outcomes across various mask types (e.g., Center, Expand, Half).

This work is the first to apply RetNet, an NLP architecture, to image completion tasks and introduces a bidirectional Bi-RetNet structure tailored for image restoration needs.

**Strengths:**

1. Achieves faster image restoration than traditional methods through pixel-wise inference.
2. Proposes a novel structure that integrates bidirectional contextual information, adapting the RetNet architecture for parallel training and recursive inference.
3. Demonstrates high restoration quality across different mask types, showing potential for real-time applications.

**Weaknesses:**

Although various mask types were tested, demonstrating performance on more complex and patterned masks and providing additional solutions could enhance the approach’s practicality in real-world settings.

**Questions:**

Further analysis or an Ablation Study is needed to clearly assess the performance gains contributed by the Bi-RetNet structure.

---

### Official Review · Reviewer_oSn5 · 2024-11-06

**Soundness:** 2
**Presentation:** 3
**Contribution:** 2
**Rating:** 3
**Confidence:** 1

**Summary:**

The paper addresses the issue of high complexity in image completion using attention mechanisms by proposing the use of an RNN-based linear attention structure—RetNet—to reduce computational complexity while enhancing performance.

**Strengths:**

1.The paper is well-structured and easy to read;

2. The contributions made by the authors are empirically validated, demonstrating the efficacy of the proposed method.

**Weaknesses:**

1. The level of innovation might be insufficient to meet the high standards of the ICLR conference, as the main contribution appears to be the application of the Ret-Net technology from NLP to the field of image completion.

2. The performance does not reach the state-of-the-art for the field, and the most recent performance comparisons in the paper are with works from 2022, which may seem outdated for a submission to ICLR 2025. The author should try to compare their conclusions with newer methods, such as [1] [2]

3. There are some apparent typos, such as the “?” on line 30, which need to be corrected to enhance the overall quality of the paper.

[1] BrushNet: A Plug-and-Play Image Inpainting Model with Decomposed Dual-Branch Diffusion ECCV 2024 (https://github.com/TencentARC/BrushNet)

[2] 'Don't Look into the Dark: Latent Codes for Pluralistic Image Inpainting CVPR 2024 (https://github.com/nintendops/latent-code-inpainting)

**Questions:**

see weaknesses

---

### Note · Authors · 2024-11-13

I have read and agree with the venue's withdrawal policy on behalf of myself and my co-authors.